# Long Non-Coding RNA PANTR1 is Associated with Poor Prognosis and Influences Angiogenesis and Apoptosis in Clear-Cell Renal Cell Cancer

**DOI:** 10.3390/cancers12051200

**Published:** 2020-05-10

**Authors:** Maximilian Seles, Georg C. Hutterer, Johannes Foßelteder, Marek Svoboda, Margit Resel, Dominik A. Barth, Renate Pichler, Thomas Bauernhofer, Richard E. Zigeuner, Karl Pummer, Ondrej Slaby, Christiane Klec, Martin Pichler

**Affiliations:** 1Department of Urology, Medical University of Graz, 8036 Graz, Austria; maximilian.seles@medunigraz.at (M.S.); georg.hutterer@medunigraz.at (G.C.H.); richard.zigeuner@medunigraz.at (R.E.Z.); karl.pummer@medunigraz.at (K.P.); 2Division of Oncology, Department of Internal Medicine, Medical University of Graz, 8036 Graz, Austria; johannes.fosselteder@medunigraz.at (J.F.); margit.resel@medunigraz.at (M.R.); dominik.barth@medunigraz.at (D.A.B.); thomas.bauernhofer@medunigraz.at (T.B.); martin.pichler@medunigraz.at (M.P.); 3“Non-coding RNAs and Genome Editing in Cancer” Research Unit, Medical University of Graz, 8036 Graz, Austria; 4Central European Institute of Technology (CEITEC), Masaryk University, 62500 Brno, Czech Republic; msvoboda@mou.cz (M.S.); ondrej.slaby@ceitec.muni.cz (O.S.); 5Department of Experimental Therapeutics, The University of Texas MD Anderson Cancer Center, Houston, TX 77030, USA; 6Department of Urology, Medical University of Innsbruck, 6020 Innsbruck, Austria; renate.pichler@i-med.ac.at; 7Department of Biology, Faculty of Medicine, Masaryk University, 62500 Brno, Czech Republic

**Keywords:** long intergenic non-coding RNA, PANTR1, Linc01158, Linc-POU3F3, siRNA, renal cell cancer, clear-cell renal cell carcinoma, oncogene

## Abstract

POU3F3 adjacent non-coding transcript 1 (PANTR1) is an oncogenic long non-coding RNA with significant influence on numerous cellular features in different types of cancer. No characterization of its role in renal cell carcinoma (RCC) is yet available. In this study, PANTR1 expression was confined to human brain and kidney tissue and was found significantly up-regulated in clear-cell renal cell carcinoma tissue (ccRCC) compared to non-cancerous kidney tissue in two independent cohorts (*p* < 0.001 for both cohorts). In uni- and multivariate Cox regression analysis, ccRCC patients with higher levels of PANTR1 showed significantly poorer disease-free survival in our own respective cohort (*n* = 175, hazard ratio: 4.3, 95% confidence interval: 1.45–12.75, *p* = 0.008) in accordance with significantly poorer overall survival in a large The Cancer Genome Atlas database (TCGA) cohort (*n* = 530, hazard ratio: 2.19, 95% confidence interval: 1.59–3.03, *p* ≤ 0.001). To study the underlying cellular mechanisms mediated by varying levels of PANTR1 in kidney cancer cells, we applied siRNA-mediated knock-down experiments in three independent ccRCC cell lines (RCC-FG, RCC-MF, 769-P). A decrease in PANTR1 levels led to significantly reduced cellular growth through activation of apoptosis in all tested cell lines. Moreover, as angiogenesis is a critical driver in ccRCC pathogenesis, we identified that PANTR1 expression is critical for in vitro tube formation and endothelial cell migration (*p* < 0.05). On the molecular level, knock-down of PANTR1 led to a decrease in Vascular Endothelial growth factor A (VEGF-A) and cell adhesion molecule laminin subunit gamma-2 (LAMC2) expression, corroborated by a positive correlation in RCC tissue (for VEGF-A *R* = 0.19, *p* < 0.0001, for LAMC2 *R* = 0.13, *p* = 0.0028). In conclusion, this study provides first evidence that PANTR1 has a relevant role in human RCC by influencing apoptosis and angiogenesis.

## 1. Introduction

Renal cell carcinoma (RCC) is still a lethal disease with rising incidence, currently representing the third most common urological malignancy, with approximately 99,000 new cases and 39,000 kidney-cancer-related deaths in the European Union in 2018 [1].

Clear-cell renal cell carcinoma (ccRCC) is a rather aggressive subtype representing two thirds of all stage RCC and 85% of metastatic RCC cases [2]. The von Hippel-Lindau/(vHL)−hypoxic inducible factor/(HIF) axis is the major carcinogenic pathway in ccRCC. Activation of this pathway leads to the upregulation of pro-angiogenic genes and therefore, can be therapeutically targeted by several anti-angiogenic drugs. In addition, this knowledge led to the recent development of specific HIF2a inhibitors in the metastatic setting of ccRCC [1,3]. At present, clinical outcome of RCC patients is predicted by multidimensional parameters and models including laboratory values, cross sectional imaging and histopathological specimen with no reliable guideline-approved biomarkers [1,4,5,6,7,8,9,10,11] Lately, intensive research has aimed to find non-operator dependent markers for individual prognosis at different stages including the pre- and post-surgery setting as well as therapeutic targets in advanced/metastatic stages of the disease [12].

Long-non-coding RNAs (lncRNAs) represent a part of low-abundant non-coding sequences of the genome with more than 200 base pairs in length [13]. The complete range of lncRNA function needs yet to be discovered and characterized but their impact at epigenetic transcriptional and post-transcriptional level seems to be substantial [14]. Aberrant functions of lncRNAs affect several hallmarks of cancer such as cell proliferation, apoptosis, angiogenesis, promotion of metastasis, and mesenchymal-to-endothelial transition [15]. Several lncRNAs with different roles in RCC carcinogenesis have been discovered but non-coding RNA research is yet in its infancy [16,17].

PANTR1 (also known as Linc-POU3F3 or LINC01158) is an intergenic lncRNA which resides 4-kb upstream from the protein-coding POU3F3 gene. POU3F3 is located on chromosome 2q12.1 and was originally discovered for its key role in neuro- and kidney development [18]. In contrast to its protein-coding neighbor, PANTR1 is furtherly known for its versatile role in carcinogenesis as a promotor of cell migration, invasion, angiogenesis and proliferation as well as a repressor of apoptosis in different types of cancer [19,20,21,22,23,24,25]. To the best of our knowledge, PANTR1 has not been comprehensively studied for its relevance in human ccRCC and for underlying cellular mechanisms.

As tumor angiogenesis is one of the most critical steps in ccRCC and PANTR1 has been associated with kidney development and tumor angiogenesis in glioma, we hypothesized that PANTR1 might play an important role in ccRCC carcinogenesis.

## 2. Results

### 2.1. PANTR1 is Up-Regulated in RCC Cancer Cells and Tissue and is Associated with Poor Survival of RCC Patients

In order to determine the relevance of PANTR1 in human kidney tissue and RCC tissue, we firstly compared the PANTR1 expression values in several normal human tissue types based on a publicly available RNAseq dataset. Interestingly, PANTR1 expression was confined to brain and kidney tissue only (Figure 1a). Relevant differences in median RNA expression could be observed for cancers belonging to these two tissue types: glioma for brain tissue and RCC for kidney tissue. Among RCCs, the clear-cell and the papillary (pRCC) but not the chromophobe (chRCC) RCC subtypes showed relevant differences in expression (Figure 1b). This observation and the fact, that ccRCC is the most common and most aggressive subtype prompted us to explore whether PANTR1 is up-regulated in ccRCC cancerous tissue compared to normal kidney tissue among representative patient cohorts. As shown in Figure 1c,d using two independent publicly available datasets, PANTR1 expression was significantly up-regulated in cancerous tissue compared to normal surrounding kidney tissue (*p* = 5,7 × 10^−10^ in cohort 1 and *p* = 0.000018 in cohort 2, Wilcoxon signed rank test). RNA in situ hybridization confirmed a specific and higher expression of PANTR1 in cancer cells of RCC tissue compared to luminal cells of non-cancerous kidney tissue (Figure 1e). Finally, to further substantiate the clinical relevance of PANTR1 in human RCC, we tested the prognostic relevance of PANTR1 in a large cohort of ccRCC patients (*n* = 170). Table 1 summarizes the clinicopathological characteristics of the cohort. The median age of the patients was 65 years (minimum: 26, maximum: 86), with a mean age of 64 (SD ± 10.8 years). Using this clinical well-annotated cohort, we measured PANTR1 expression by qRT-PCR in these samples. As shown in Figure 1f, high levels of PANTR1 expression was associated with significantly shorter recurrence-free survival (*p* = 0.045). Adding PANTR1 into a multivariate Cox model including well-known prognostic factors such as age, gender, tumor grading and stage, PANTR1 prevailed as an independent prognostic factor (hazard ratio: 4.3, 95% confidence interval: 1.45–12.75, *p* = 0.008, Table 2). In order to confirm the prognostic value of PANTR1 in an independent cohort, we analyzed data of the Cancer Genome Atlas for ccRCC cases (*n* = 530). As shown in Appendix A, high PANTR1 expression prevailed as a poor prognostic factor for overall survival in this external cohort (hazard ration: 2.19, 95% confidence interval: 1.59–3.03, *p* < 0.001). Thus, based on the findings that PANTR1 is up-regulated in RCC tissue and its association with a higher risk of disease-recurrence, we aimed to characterize possible cellular and molecular mechanisms.

### 2.2. Reduced Levels of PANTR1 Decrease Cellular Growth

These initial promising results encouraged us to further characterize the biological role of PANTR1 in ccRCC. As a first step, we verified its expression in three ccRCC cell lines (RCC-FG, RCC-MF and 769-P) and in human umbilical vein endothelial cells (HUVECs) (Appendix A). Next, we employed a transient knock-down approach using two independent siRNAs (Appendix A). We then explored the effect of PANTR1 knock-down on cellular growth in the three independent ccRCC cell lines. In all ccRCC cancer cell lines, cellular growth was significantly reduced in cells with decreased PANTR1 expression levels compared to control conditions (Figure 2a–c).

### 2.3. PANTR1 Silencing Induces Apoptosis

After confirmation of reduced cellular growth in ccRCC cell lines with decreased PANTR1 expression, we further explored whether apoptosis is the mode of cellular action for the reduced cellular growth pattern. Reduced levels of PANTR1 expression led to an increased activity of caspase 3/7 after 72 and 96 h compared to control conditions (Figure 3a–c). To confirm the increased apoptotic activity with a second independent method, western blot analysis confirmed an increased cleavage of PARP1 in PANTR1 knocked-down cells (89 kDa band, which is a marker for increased apoptosis [26]) (Figure 3d–f).

### 2.4. PANTR1 Knock-Down Decreases Key Parameters of Angiogenesis and Endothelial Cell Migration

Based on the previously reported results of PANTR1 regarding its angiogenic potential in glioma [23], we used in vitro angiogenesis models to test the hypothesis that PANTR1 influences angiogenesis. We firstly confirmed the expression (Appendix A) as well as the successful siRNA-mediated knock down of PANTR1 (Appendix A) in a cell line model system for endothelial cells i.e., human umbilical vein endothelial cells (HUVECs). After performing a tube formation assay and by consecutive analysis and comparison of several key parameters involved in tube formation, we revealed that PANTR1 knock-down led to a significantly reduced capacity of tube formation in comparison to control conditions (Figure 4a–e). As migration of endothelial cells is another important step in formation of new blood vessels and therefore also essential for angiogenesis in the context of tumor formation [27], we additionally conducted an endothelial cell specific migration assay. Knock-down of PANTR1 resulted in reduced cellular migration of HUVECs when compared to control conditions (Figure 4f). To explore whether PANTR1 influences angiogenesis-related factors in ccRCC cancer cells, we measured the gene expression levels of several angiogenetic factors upon PANTR1 knock-down. Overall measuring a mRNA panel of eight angiogenesis-related genes, a significant down-regulation of the pro-angiogenic gene VEGF-A [28] and migration associated protein LAMC2 [29] was observed for both independent siRNAs (Figure 4g). These cell line findings are corroborated by a positive (though weak) correlation of PANTR1 with VEGF-A (*R* = 0.19, *p* < 0.001) and LAMC2 in RCC cancer tissue (*R* = 0.13, *p* = 0.0028) (Figure 4h,i).

## 3. Discussion

In this study, we report for the first time the involvement of the long intergenic non-coding RNA PANTR1 in ccRCC carcinogenesis. LincRNA PANTR1 (long intergenic non-protein coding RNA 1158/Linc01158, Linc-POU3F3) is a non-protein coding transcript with four variants about 4-kb upstream of the POU3F3 gene on chromosome 2q12.1 [30]. POU3F3 is a well-known key player in development of the human central nervous system and the kidneys. POU3F3 homozygous knockout mice display highly increased plasma parameters of decreased kidney function with accompanying developmental defects in the forebrain and hippocampus. Thorough examinations reveal smaller kidneys containing dysmorphic loops of Henley, distal convoluted tubule (DCT), macula densa and proximal tubules although this was difficult to determine [31]. Humans with mutations of the POU3F3 gene display a wide range of features directly linked to the size of the mutation spanning from mild cognitive impairment and mild dysmorphosis to complex malformation of the kidney, corpus callosum, heart and anus [32,33]. Relevant expression is found in human brain tissue and kidney tissue with a peak at week 10 during fetal development with later decrease. This implicates a supporting role for POU3F3 in normal human kidneys. We could observe the same pattern of RNA expression for PANTR1 in normal brain and kidney tissue only. Interestingly, we observed higher expression levels of PANTR1 in ccRCC and pRCC compared to corresponding normal kidney tissue, whereas lower (or even non-detectable) PANTR1 expression in chRCC cases. The reasons for this observation remain speculative, but different underlying pathophysiology might reflect the varying expression pattern.

ccRCC is regarded as the prototype of RCC when it comes to angiogenesis. This is reflected by the central role of the vHL-HIF–pathway. In ccRCC, frequent loss of chromosome 3p occurs which includes deletion of 3p25 where the vHL gene resides [34,35]. Mutated vHL leads to several inappropriate functions of the cell, the most relevant being the activation of HIF subunits by polyubiquitination. In the case of ccRCC, HIF escapes the recognition of mutated vHL protein, dimerizes with HIF-Beta leading to activation of around 200+ genes which normally promote responses to low oxygen environment such as VEGF, PDGF, EGFR, c-met and cyclin D [36], resulting in increased angiogenesis. In non-cancerous conditions such as the von-Hippel-Lindau-syndrome, mutated vHL protein also causes angiogenesis-dependent tumors including haemangioblastomas, a tumor arising from blood vessel in the central nervous system [35]. pRCC comprises two different subtypes, type I and type II which are clinically and biologically different [37]. Especially, in type I MET pathway is frequently altered, which plays an important role in angiogenesis in analogy to ccRCC. On the other hand, chRCC is completely different from a cellular point of view. Here, driving mutations are losses of chromosomes Y, 1, 2, 6, 10, 13, 17 and 21 [34]. Angiogenesis seems not to play that important role which is also supported by the notion of modest effects of multi-kinase inhibitors aiming for vHL downstream targets in the clinical setting [38]. So, the differences in underlying genetics and pathophysiology including angiogenesis may be one factor that could explain the different patterns of RNA expression for PANTR1 in ccRCC and pRCC in contrast to chRCC.

So far, several studies have identified PANTR1 as an oncogenic driver in different types of cancer such as esophageal squamous cell cancer (eSCC) [39], glioma [23,25], hepatocellular carcinoma [20], cervical [40], colorectal [21], gastric [41], prostate [24] and breast cancer [42]. In these studies, PANTR1 has been shown to promote angiogenesis, cell proliferation, migration and invasion and to inhibit apoptosis. Additionally, and in accordance with our ccRCC cohort, PANTR1 alone as well as in combination with other lncRNAs could identify patients with early progress in eSCC, breast and cervical cancer [39,40,42]. Further correlation with different clinicopathological parameters could be demonstrated in cervical and colorectal cancer in concordance with our own data [21,40]. In our study, we found significantly higher expression levels of PANTR1 in ccRCC tissue compared to normal kidney tissue and that high PANTR1 expression is associated with poor clinical outcome of patients in two independent cohorts.

Driven by these relevant results in human cancer samples, we wanted to set the clinical findings in a biological context of ccRCC cells with regard to certain cellular features crucially contributing to carcinogenesis [43]. In a first step, we investigated the influence of PANTR1 on the probably most important trait of cancer cells—their ability of a sustained proliferation. This unique characteristic allows cancer cells to survive beyond the normal cellular lifespan and to proliferate abnormally. We could demonstrate that knock-down of PANTR1 leads to significantly reduced cellular growth in all three ccRCC cell lines. The observed effects size of cellular growth and apoptosis varied in the three different ccRCC cell lines, which probably reflects genetic and molecular heterogeneity encountered also in human RCC [44]. For instance, drugs such as anti-angiogenic drugs are only working in about one third of ccRCC patients, and effects on tumor shrinkages substantially varies between patients [45].

In order to clarify the mode of cellular action observed by reduced cellular growth, we investigated apoptotic activity after PANTR1 knock-down. One form of programed cell death, also known as apoptosis, is a natural regulator to overcome indefinite life of (cancer) cells [43]. Pro-apoptotic activity of PANTR1 knock-down was demonstrated by significantly elevated activities of caspases 3 and 7 after PANTR1 knock-down. Caspases 3 and 7 represent two of the 12 caspases in the cascade of apoptosis and are responsible for direct execution of apoptosis in humans [46]. This pro-apoptotic phenotype was further confirmed by measuring cleaved PARP1 protein expression, an additional marker for apoptosis induction [47] by Western Blot analysis after PANTR1 knock-down.

Faster proliferation, longer cell survival and ineffective metabolism of malignant tumors end up with the need for faster delivery of nutrients and oxygen. The tumor starts to build new pipelines, a process called angiogenesis which marks a hallmark of cancer. PANTR1 apparently plays a role in angiogenesis in other types of cancer [23,25]. Using a tube formation assay in HUVECs we could demonstrate that PANTR1 is highly capable of enforcing this process of constructing new vessels to improve the tumor’s supply chain. The expression of VEGF-A, a major growth factor downstream of vHL and HIF [48], was significantly decreased after PANTR1 knock-down as a further proof that angiogenesis is regulated on a multilayered process by PANTR1.

Blood vessel formation needs the migration of endothelial cells. By using an endothelial cell migration assay, we identified that this process is disturbed by low PANTR1 expression levels. Moreover, after knock-down of PANTR1 we could further demonstrate a significant decrease of LAMC2 mRNA expression in ccRCC cells. LAMC2 is a gene encoding for a subunit of an essential component of epithelial basement membranes named glycoprotein laminin-332 which is regulating cell motility and adhesion [49]. It has been shown to inhibit angiogenesis in cholangiocarcinoma [50]—a fact further strengthening a contribution of PANTR1 to angiogenesis. Increase of LAMC2 has been shown in different kinds of cancers [49,50,51,52] to further underline its role in carcinogenesis.

Our study identified the lncRNA PANTR1 as potential molecular marker for ccRCC prognosis and we could demonstrate an involvement in renal carcinogenesis. As already mentioned, further studies unraveling the underlying molecular mechanism are needed but we speculate that PANTR1 could exert a function as transcriptional regulator of gene expression of carcinogenesis-associated genes. LncRNAs have been demonstrated to act as epigenetic regulators of transcription in cancer by exerting different functions such as i) guide lncRNA—by recruiting or rejecting epigenetic regulators onto specific chromosomal loci; ii) architect lncRNAs—by modifying the three-dimensional conformation of chromatin or iii) enhancer lncRNAs—by regulating transcription through enhancer like functions [53,54]. Some representative examples are the epigenetic regulation of the INK4b/ARF/INK4a locus via the lncRNA Antisense Noncoding RNA In The INK4 Locus (ANRIL) [55], the upregulation of NKD inhibitor of WNT signaling pathway 1 (NKD1) transcription by lncRNA H19 via NKD inhibitor of WNT signaling pathway 1 (EZH2) [56] or HOX Transcript Antisense RNA (HOTAIR)-mediated repression of polycomb repressive complex 2 (PRC2) transcription. [57] Although solid experimental evidence is partially lacking, also PANTR1 is speculated to epigenetically regulate the expression of ROCK1 in prostate cancer [24], of TGF-β1 in nasopharyngeal and gastric cancer. [19,41], of several angiogenesis-related genes and corresponding proteins such as bFGF, bFGFR, VEGF-A, and Angio in glioma [23] and of the miR-449a/KLF4 pathway in cardiac in-stent restenosis [58].

Overall, our data show for the first time that PANTR1 plays a relevant role in human kidney cancer which can be partly explained by its influence on apoptosis and tumor angiogenesis. Additionally, PANTR1 is capable of predicting disease-free and overall survival in RCC patients. Further studies are now needed to clarify its role in other subtypes of RCC [59], to explore the pathways it is involved in and to eventually discover therapeutics for which PANTR1 could serve as a target.

## 4. Materials and Methods

### 4.1. Analysis of Gene Expression and Survival Data

Tissue-specific expression data was derived from a publicly available database [60] originally issued in a publication by Fagerberg et al. [61]. To assess expression differences of PANTR1 between normal kidney tissue and ccRCC tissue, data from the gene expression omnibus (GEO) was analyzed. For cohort 1, we derived data from Roemeling et al. [62] who previously measured the expression of thousands of genes in ccRCC and normal kidney tissue of the same patients by micro arrays (GPL570). They published their data as open access with the GEO accession number GSE53757 [63]. The dataset includes 144 samples, which are paired malignant and normal kidney tissues from 72 patients. The data was analyzed using the open-source statistics software “R” version 3.3.3 (The R Foundation, Boston, MA, USA) with the Bioconductor extension (https://www.bioconductor.org). The software was then used to create box-plots that compare PANTR1 expression in malignant and healthy tissue. Wilcoxon signed rank test was used to assess expression differences between matched ccRCC and healthy kidney samples. For cohort 2, The Cancer Genome Atlas (TCGA) database was used to obtain freely available expression data. The data analysis was restricted to ccRCC patients, which resulted in a cohort of 434 patients. The data was then diminished for expression values of PANTR1 and analyzed with R software. For correlation analysis between PANTR1 and other genes, we used the TCGA PanCancer Atlas data (*n* = 512) [64]. For calculating the prognostic relevance of PANTR1, we used an external dataset in connection with real-patient tissue samples (*n* = 170) provided by Ondrej Slaby from the Masaryk University (Brno, Czech Republic). Written informed consent was obtained from all patients, and the study was approved by the local Ethics Board at the Masaryk Memorial Cancer Institute. All experiments were performed in accordance with relevant guidelines and regulations. RNA was isolated from fresh-frozen tissue sections. Expression values of PANTR1 were measured by qRT-PCR and were normalized to the housekeeping gene Peptidylprolyl Isomerase A by ΔCT-method and log2 transformed. Corresponding clinic-pathological data from the Department of Comprehensive Cancer Care, Masaryk Memorial Cancer Institute, Czech Republic of these patients between 25 to 90 years of age with an initial diagnose between 2004 to 2013 was obtained. Patients were treated by standard surgical procedures and received further treatment when indicated. Patients were stratified into a low and a high expression group according to a cut off value calculated by ROC analysis. Disease-free (DFS) survival probabilities were estimated using the Kaplan-Meier method and multivariate Cox proportional analysis in SPSS (IBM Corp., Armonk, NY, USA). For confirmation of the prognostic value of PANTR1, we used publicly available RNAseq data from the TCGA Pan-Cancer Set derived by Kaplan Meier plotter [65].

### 4.2. In Situ Hybridization

Sections of FFPE tissue (thickness 4 µm) were mounted on Superfrost Plus coated slides (Thermo Scientific, Vienna, Austria) and the slides were processed according to manufacturer’s instructions for the RNAscope 2.0 High Definition—BROWN kit (Advanced Cell Diagnostics Inc., Newark, CA, USA). Three sections of each sample were hybridized with the following probes: Hs-LINC1158 (#514801, i.e., PANTR1), negative control *DapB* (#310043) and positive control *Hs-PPIB* (#313901). For image analysis, a representative tumor region was selected for each tumor and images were captured at 40× magnification using an Observer.Z1 inverted microscope (Zeiss, Oberkochen, Germany). *DapB* and *PPIB* probes served as technical quality controls that needed to fulfil the cut-off criteria (≤0.5 spots/cell for negative controls; ≥2.5 spots/cell for positive controls) in order to ensure technical specificity of the probes and to detect samples with highly degraded RNA.

### 4.3. Cell Culture

Human clear-cell RCC cell lines RCC-FG1/KTCTL-26, RCC-MF/KTCTL-1M (both from Cell Line Service (CLS), Eppelheim, Germany) and 769-P (from ATCC; American Type Culture Collection USA) were cultured in RPMI1640 growth media (Gibco, ThermoFischer Scientific, Vienna, Austria) containing 10% fetal bovine serum (Hyclone, GE Healthcare, Frankfurt/Main, Germany) and 1% penicillin/streptomycin mixture (Gibco). Human umbilical vein endothelial cells (HUVECs) were cultured in EBM™-2-Medium with necessary supplements (EGM™-2 SingleQuots™; all purchased from Lonza, Basel, Switzerland). All cell lines were kept at 37 °C in a humidified 5% CO_2_ atmosphere.

### 4.4. Transient Transfection with siRNAs

All three cell lines were transiently transfected using a final concentration of 50 nM of two different siRNAs targeting PANTR1 (siRNA 1 #n507582, siRNA 2 #n507583; Ambion Silencer Select, Ambion, Austin, TX, USA) according to the fast forward protocol of HiPerFect Transfection Reagent (Qiagen, Hilden, Germany). Non-targeting negative control siRNA (Silencer Select negative Control No.1 siRNA #4390843; Ambion Silencer Select, Ambion, Austin, TX, USA) and cell death control (AllStars Hs Cell Death siRNA # SI04381048, Qiagen, Hilden, Germany) served as references.

### 4.5. RNA Isolation and qRT-PCR

For detection of PANTR1 and other gene expression levels in ccRCC cells (confluency of approximately 75–90%) and human tissue, total RNA was isolated using a standard Trizol (Invitrogen, Carlsbad, CA, USA) protocol according to the manufacturer’s instructions. RNA was stored at −80 °C until further processing. cDNA was synthesized from 1 µg of total RNA using the QuantiTect Reverse Transcription Kit (Qiagen, Venlo, The Netherlands). Quantification of lncRNAs and mRNAs was performed using the QuantiTect SYBR Green PCR kit (Qiagen) according to the manufacturer’s recommendations on a LightCycler 480 real-time PCR system (Roche, Mannheim, Germany). U6 (Eurofins Scientific, Vienna, Austria) was used for normalization of lncRNAs and GAPDH (Eurofins) for protein coding genes. Differences in fold expression were calculated in Excel (Microsoft Corporation, Redmond, WA, USA) following the 2^−ΔΔCt^ method. All specific primer sequences for analyzed genes are listed in Appendix A. Detailed experimental performance according to MIQE guidelines are provided in Appendix A.

### 4.6. Caspase 3/7 Apoptosis Assay

Activity of caspases 3 and 7 was determined by Caspase-Glo 3/7 assay (Promega, Madison, WI, USA) according to the manufacturer’s instructions. All three ccRCC cell lines were seeded in 96-well plates and transfected with the two different siRNAs against PANTR1 and the controls as described above. Luminescence was recorded using white 96-well plates (Thermo Scientific) and a luminometer (LumiStar, BMG Labtech, Ortenberg, Germany) 72 and 96 h after transfection. All experiments were performed in four independent repeats. Differences in luminescence between the control condition and the two corresponding transfected cell lines were calculated using the ANOVA method.

### 4.7. Western Blot

Total protein was extracted with radioimmunoprecipitation assay (RIPA) buffer (150 mM NaCl, 50 mM Tris-HCl, pH 7.5, 1% Triton, 0.1% SDS, 0.1% sodium deoxycholate and 1% Nonidet P40). 25 µg of total cellular proteins were resuspended in laemmli buffer (4% SDS, 20% glycerol, 10% 2-mercaptoethanol, 0.004% bromphenol blue and 0.125 M Tris HCl, pH approx. 6.8) and heated at 95 °C for 5 min. Proteins were separated by a 4–15% Mini-PROTEAN^®^ TGX™Precast Gel (BioRad, Hercules, CA, USA) and transferred onto a nitrocellulose membrane (BioRad). The membrane was blocked for one hour with 5% non-fat dry milk in Tris buffered Saline/0.1% Tween-20. Immunoblotting was performed and antibodies specific for PARP (Cell Signaling, Danvers, MA, USA diluted 1:1000 in 5% non-fat dry milk in Tris buffered Saline/0.1% Tween-20) and Cofilin as loading control (Cell Signaling, diluted 1:2000 in 5% non-fat dry milk in Tris buffered Saline/0.1% Tween-20) were detected using a horseradish peroxidase (HRP)-conjugated anti-rabbit antibody (Santa Cruz Biotechnology Inc., Dallas, TX, USA). Cells treated for three hours with 1 µM staurosporin were used as apoptosis-induction (positive) control. Visualization was performed using an enhanced chemiluminescence detection system (Super Signal West Pico, Thermo Scientific, Rockford, IL, SUA) on a ChemiDoc^TM^ Imaging System (BioRad). Densitometric analysis was performed with ImageJ software.

### 4.8. WST-1 Proliferation Assay

For the WST-1 proliferation assay (Roche) 5 × 10^3^ cells (RCC-MF, RCC-FG) or 3 × 10^3^ cells (769-P) were seeded into 96-well culture plates and transfected with a final concentration of 50 nM siRNAs according to manufacturer instructions. Cells were incubated with 200 µL growth medium for 96 h and every 24 h WST-1 reagent was applied. Colorimetric changes were measured with a spectrophotometer (SPECTROStar Omega, BMG Labtech, Ortenberg, Germany) at 450 nm with a reference wavelength of 620 nm. Differences between the control condition and the two corresponding transfected cell lines were calculated using the ANOVA method.

### 4.9. Tube Formation Assay

For the tube formation assay (In vitro Angiogenesis Assay kit ECM625, Merck, Vienna, Austria) HUVECs were transfected in 6-well plates as described above for 48 h. Wells of 96-well plate were covered with 50 µL ECMatrix^TM^ before 1.5 × 10^4^ transfected HUVECs were seeded on top. Cells were incubated at 37 °C and 5% CO_2_ for 16 h. Tube formation was using an inverted light microscope (Olympus IX71, Tokyo, Japan). Images of three randomly selected areas were used for quantification. Exemplary parameters such as number of junctions, nodes, meshes and branches as well as tube length were analyzed using the ImageJ plugin “Angiogenesis Analyzer” (U. S. National Institutes of Health, Bethesda, MD, USA).

### 4.10. Endothelial Migration Assay

Endothelial cell migration was assessed with the QCM 3 µm Endothelial Cell Migration Assay Fibronectin (ECM200, Merck, Darmstadt, Germany). Therefore, 3 × 10^5^ HUVECs were seeded on 60 mm dishes and transfected with a final concentration of 50 nM siRNA (negative control or targeting PANTR1) for 48 h. Migration assay was conducted according to manufacturer’s protocol. Colorimetric changes were recorded on a spectrophotometer (SPECTROStar Omega, BMG Labtech, Ortenberg, Germany) at a wavelength of 570 nm. Differences between the control condition and the two corresponding transfected cell lines were calculated using the ANOVA method.

### 4.11. Statistical Analysis

All experiments were performed at least three times. Mean values and standard deviation (SD) were calculated and indicated. *p* values below 0.05 were regarded as significant and marked by an asterisk (*). *p* values < 0.01 and <0.001 were asterisked when applicable with (**) and (***), respectively. Unless otherwise stated, calculations were performed with Prism 5.0 (GraphPad Software Inc., San Diego, CA, USA). If applicable, other statistical methods and programs that were used are annotated in the corresponding paragraphs.

## 5. Conclusions

The major role of PANTR1 in renal development and the remarkable impact on different hallmarks of cancer lead us to the hypothesis of PANTR1 playing a noteworthy role in ccRCC carcinogenesis. Long non-coding intergenic RNA PANTR1 could serve as a target for further investigation towards a better understanding of RCC and its rapidly emerging therapy strategies.

## Figures and Tables

**Figure 1 cancers-12-01200-f001:**
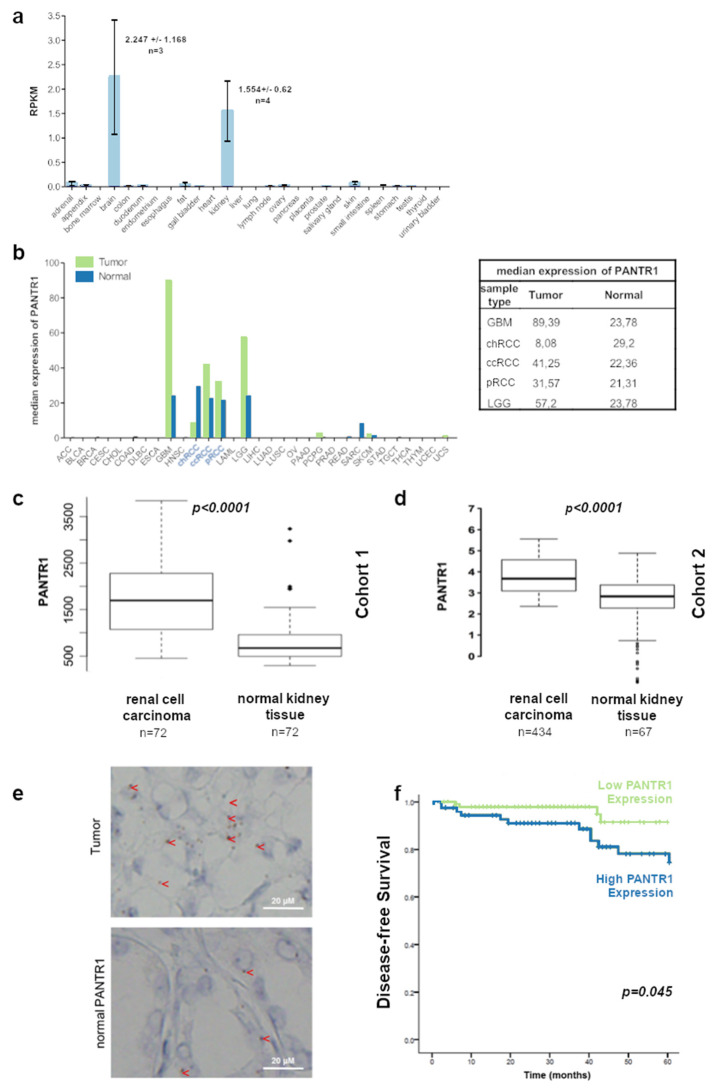
PANTR1 expression is increased in clear-cell renal cell carcinoma (ccRCC) and is associated with significantly shorter recurrence-free survival. (**a**) RNA-sequencing data showing the tissue specific expression of PANTR1 of normal samples from 95 human individuals representing 27 different tissues. Data were derived from a publicly available database (project ID: PRJEB4337, https://www.ncbi.nlm.nih.gov/bioproject/?term=PRJEB4337). (**b**) Gene expression profile of PANTR1 across tumor samples (green) and paired normal tissue (blue). The height of the bars represents the median expression. Data were derived from the publicly available Gepia-server. GBM: glioblastoma multiforme; chRCC: chromophobe RCC; ccRCC: clear-cell RCC; pRCC: papillary RCC; LGG: brain lower grade glioma. (**c**,**d**) PANTR1 expression in ccRCC versus normal kidney tissue of two different cohorts indicate higher expression levels in cancerous tissue. (**e**) Representative images of PANTR1 RNA in-situ hybridization on kidney tumor tissue (upper panel) and normal kidney tissue (lower panel). The brown dots representing PANTR1 signal mainly in the nucleus and are indicated by a red arrow head. (**f**) Kaplan-Meier plot comparing 5-year disease-free survival of ccRCC patients stratified by PANTR1 expression (low in green vs. high in blue, *n* = 175, *p* = 0.045).

**Figure 2 cancers-12-01200-f002:**
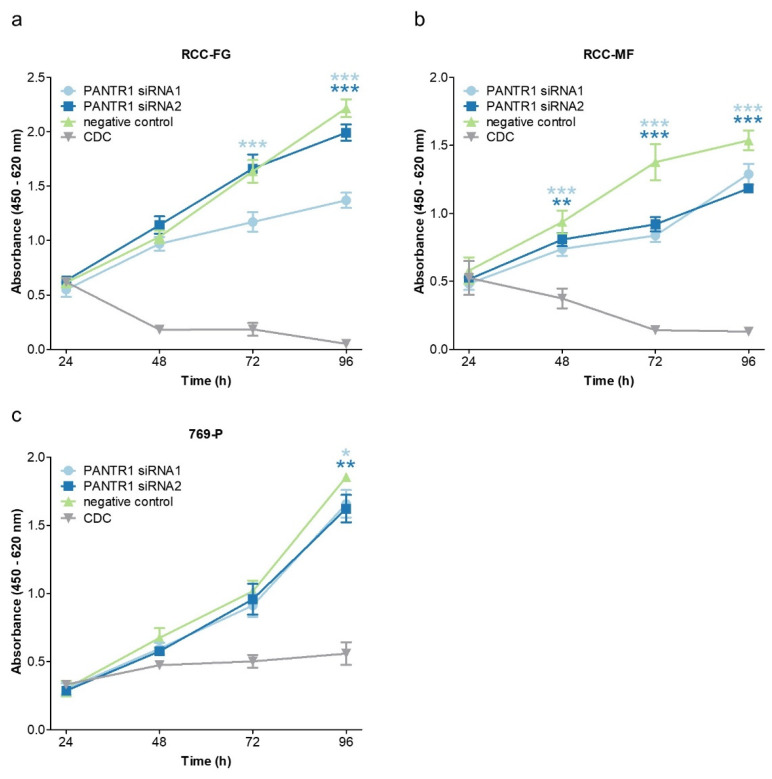
PANTR1 silencing reduces cellular growth in RCC cells. (**a**–**c**) WST-1 proliferation assay in RCC cell lines RCC-FG, RCC-MF and 769-P over 96 h either under control conditions (negative control siRNA) or after siRNA-mediated PANTR1 knock-down with two independent siRNAs. Cell death control (CDC) siRNA was used as positive control for cell growth reduction; *n* = 6. * *p* < 0.05, ** *p* < 0.01, *** *p* < 0.001.

**Figure 3 cancers-12-01200-f003:**
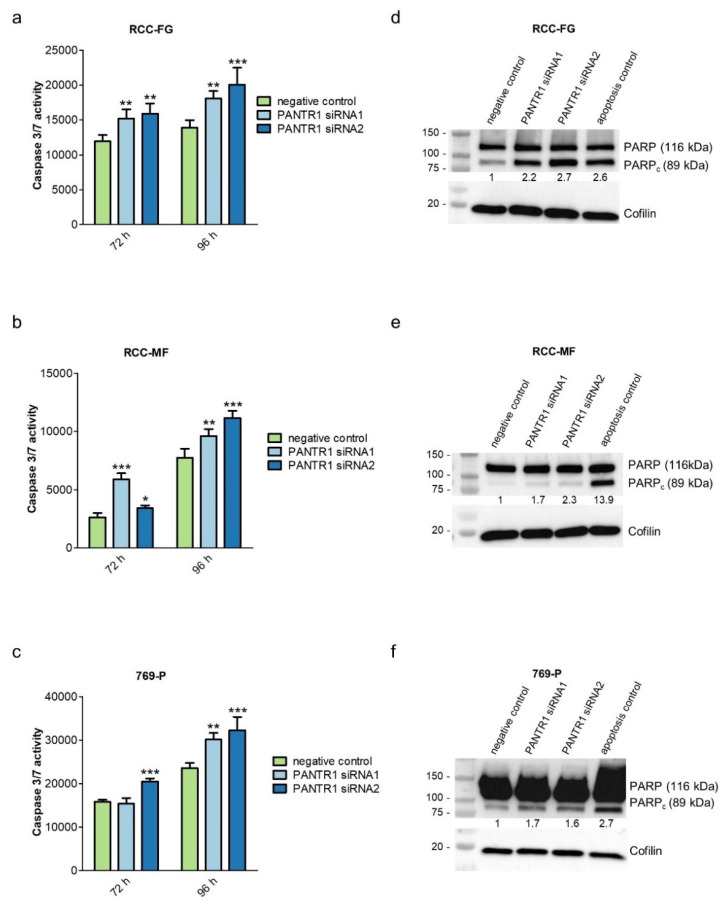
PANTR1 knock-down triggers induction of apoptosis. (**a**–**c**) Caspase 3/7 assay either under control conditions (negative control siRNA) or after siRNA-mediated knock-down of PANTR1 72 or 96 h after transfection; *n* = 4; mean ± SD. * *p* < 0.05, ** *p* < 0.01, *** *p* < 0.001. (**d**–**f**) Results of Western Blot analysis of PARP staining showing levels of full-length PARP (116 kDa) and cleaved PARP (89 kDa) in PANTR1 silenced cells 96 h post-transfection compared to control cells. Cofilin was used as loading control. Values between the blots represent the densitometric analysis of the cleaved PARP1 bands (PARP_c_, 89 kDa) normalized to the housekeeping protein cofilin. The negative control band was set as reference (=1). Uncropped images can be found in Appendix A.

**Figure 4 cancers-12-01200-f004:**
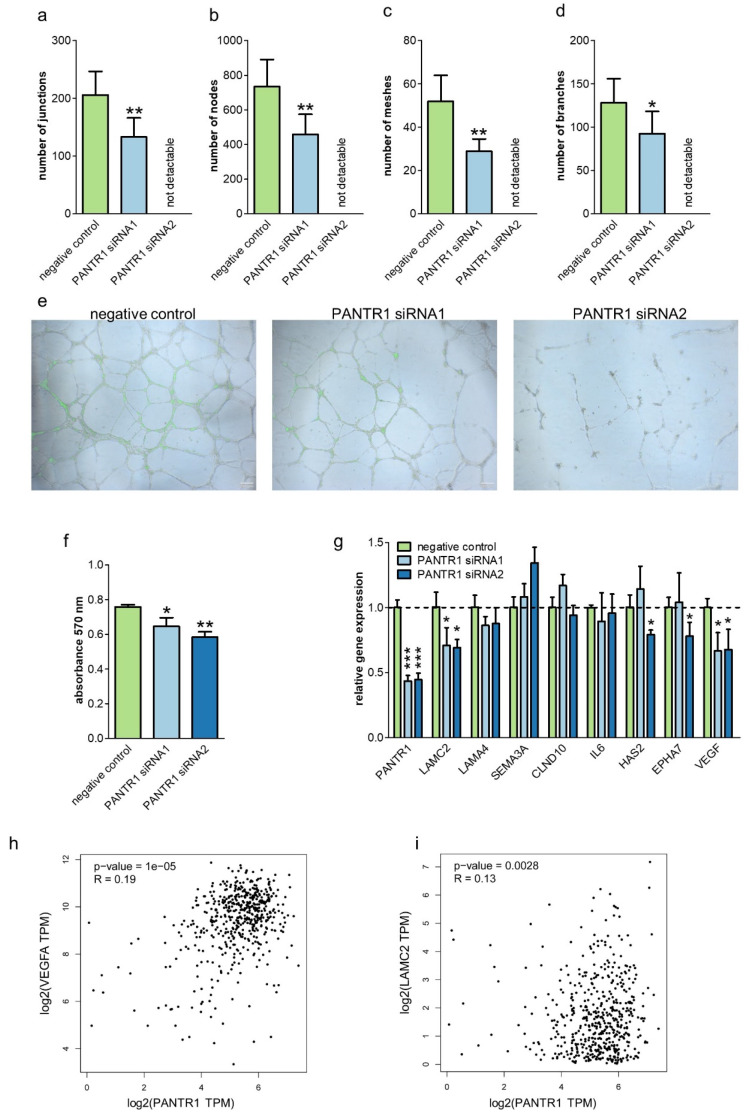
PANTR1 knock-down reduces endothelial tube formation, expression of angiogenic genes and endothelial cell migration. (**a**–**d**) Evaluation of key features of tube formation either under control conditions (negative control siRNA) or after PANTR1 knock-down; mean ± SD. (**e**) Corresponding representative pictures of tube formation assay. (**f**) Endothelial cell migration of HUVECS under control conditions (negative control siRNA) or after PANTR1 silencing; *n* = 3. Mean ± SD. * *p* < 0.05, ** *p* < 0.01, *** *p* < 0.001. (**g**) Quantification of mRNA levels of the angiogenesis-associated genes after PANTR1 knock-down in RCC-FG cells; *n* = 3. (**h**) Correlation analysis of PANTR1 and VEGF-A in ccRCC samples of the Cancer Genome Atlas. Data was derived from the publicly available gepia server. *R* = 0.19, *p* = 1.0^−5^. (**i**) Correlation analysis of PANTR1 and LAMC2 in ccRCC samples of the Cancer Genome Atlas (*R* = 0.13, *p* = 0.0028). Data was derived from the publicly available gepia server.

**Table 1 cancers-12-01200-t001:** Clinicopathological parameters of ccRCC patients in the study cohort (*n* = 175).

Parameter	Number of Patients	%
Total	175	100
Gender		
Male	108	61.7
Female	67	38.3
Grading		
G1	37	21.1
G2	93	53.1
G3	37	21.1
G4	7	4
GX	1	0.6
UICC TNM Stage		
I	124	70.9
II	22	12.6
III	27	15.4
IV	2	1.1
Pathologic T stage		
pT1	121	69.1
pT2	23	13.1
pT3	28	16.0
pT4	2	1.1
N/A	1	0.6
Lymph node metastasis		
Yes	2	1.1
No	173	98.9
Disease Recurrence		
Yes	17	9.7
No	158	90.3

**Table 2 cancers-12-01200-t002:** Multivariate Cox regression model of clinicopathological parameters and PANTR1 levels for the prediction of recurrence-free survival in ccRCC patients. Asterix (*) indicates *p*-values < 0.05. Abbreviations: CI = confidence interval; HR = hazard ratio.

Multivariate Analysis		
Parameter	HR (95% CI)	*p*-Value
Age (continuous)	1.02 (0.96–1.08)	0.341
Gender (female versus male)	0.66 (0.23–1.85)	0.428
Tumor Grade (G1+2 versus G3+4)	1.68 (0.94–2.99)	0.079
UICC TNM Stage (I/II versus III/IV	2.38 (1.35–4.19)	0.003 *
PANTR1 expression (low versus high group)	4.30 (1.45–12.76)	0.008 *

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
