# Peer review of "Long Non-Coding RNA PANTR1 is Associated with Poor Prognosis and Influences Angiogenesis and Apoptosis in Clear-Cell Renal Cell Cancer"

_cancers, 2020, doi:10.3390/cancers12051200_

Round 1

Reviewer 1 Report

The authors provide an interesting and well written report on the lncRNA PANTR1 in renal celll carcinoma. I have only a few comments.

  • Survival Analyses: (1) In order to increase the meaningfulness of the survival analysis, I suggest to include Kaplan Meier / Cox Regression analyses for the datasets GSE53757 and TCGA data sets, especially analysing survival data in the TCGA dataset would be very helpful to validate the Brno dataset. (2) Grading and Age should be reported in Table 1. In the multivariate analysis, it is reported that "stage" was analysed, as Table 1 reports multiple staging systems, which one was used? pT-stage UICC stage? please specify. (3) Did the authors use PANTR1 expression as continuous or categorised variable in the multivariate analyses? The same is for stage/grade, I think the analysis has to be critically reviewed, the Table in its present form seems to be inaccurate. (4) It would be interesting to include Overall/Cancer-specific survival in the analyses.
  • Fig. 1c/d: I think reporting p<0.0001 makes more sense than p=0.000018/p=5,7x10-10.
  • Reporting of QPCR experiments could be done according the MIQE guidelines.

Author Response

Response to Reviewer 1:

Comments and Suggestions for Authors

Reviewer # 1: The authors provide an interesting and well written report on the lncRNA PANTR1 in renal cell carcinoma. I have only a few comments.

Authors reply: We thank the reviewer for this fair review process and also the appreciation of “interesting and well written report” of our manuscript.

Reviewer #1: Survival Analyses: (1) In order to increase the meaningfulness of the survival analysis, I suggest to include Kaplan Meier / Cox Regression analyses for the datasets GSE53757 and TCGA data sets, especially analysing survival data in the TCGA dataset would be very helpful to validate the Brno dataset.

Authors reply: In order to follow the reviewers suggestion we analyzed the GSE53757 and TCGA dataset for prognostic purposes. For the GEO53757, unfortunately, no survival data is available. However, TCGA Pan-Cancer Dataset provided us the information for 530 patients where we could confirm that high PANTR1 expression also represents as a poor prognostic factor for overall survival (hazard ration: 2.19, 95% confidence interval: 1.59-3.03, p<0.001). We added this information to the result section and a new Supplementary Figure 1.

Reviewer #1: Grading and Age should be reported in Table 1.

Authors reply: We added now the information about tumor grading in Table 1 and report age distribution in the results section as it reads now as follows: “The median age of the patients was 65 years (minimum: 26, maximum: 86), with a mean age of 64 (SD ± 10.8 years).”

Reviewer #1: In the multivariate analysis, it is reported that "stage" was analyzed, as Table 1 reports multiple staging systems, which one was used? pT-stage UICC stage? please specify.

Authors reply: We added the information to Table 2, as we used the UICC TNM stage.

Reviewer #1: Did the authors use PANTR1 expression as continuous or categorized variable in the multivariate analyses? The same is for stage/grade, I think the analysis has to be critically reviewed, the Table in its present form seems to be inaccurate.

Authors reply: We have now modified the Table 2 for accurate information according to the reviewer´s suggestions.

Reviewer #1: It would be interesting to include Overall/Cancer-specific survival in the analyses.

Authors reply: To pick up the reviewer suggestion, we now included a second independent cohort of the TCGA dataset with an analysis of a further endpoint: overall survival (Supplementary Figure 1) We could confirm that PANTR1 expression is a negative prognostic factor in clear cell RCC patients (for our own cohort of 175 patients, only disease-free survival data was available, so we could not analyze OS or CSS data).

Reviewer #1: Fig. 1c/d: I think reporting p<0.0001 makes more sense than p=0.000018/p=5,7x10-10.

Authors reply: We have changed the reporting of the p-value according to the reviewer’s suggestion.

Reviewer #1: Reporting of QPCR experiments could be done according the MIQE guidelines.

Authors reply: We have included specific information concerning the qPCR experiments according to MIQE guidelines in Appendix A.

Reviewer 2 Report

This manuscript represents a simple and straightforward study of potential roles of non-coding RNAs in cancer progression. Specifically, the study aimed to identify a molecular marker associated with clear-cell renal cell cancer prognosis. The authors claim that they identified such a marker, lncRNA PANTR1. The field is still in a stage of data accumulation, so the more markers are found the better our chances to develop new targeted cancer treatment in future.

Overall it is a reasonably well-written manuscript. Yet, it has some grammar issues and wrong wording throughout the text. Discussion section in its present form is a rephrased version of Results. If the authors have very little to discuss they should consider combining Results and Discussion.

My major concern is that it seems the authors overstate the significance of some of their findings. In abstract (line 40) and Results (line 177) they are talking about ‘significant correlation’. What is this significant correlation? Correlation could be weak or strong, positive or negative. The calculated in this study correlation coefficients R=0.19 and 0.13, while being statistically significant, indicate very weak positive correlation.

Effects of PANTR1 knock-down on cell growth and caspase activity are rather minor and somewhat inconsistent (Figures 2 and 3). The fact that changes were statistically significant, does not make the changes prominent.

Some minor points and suggestions for further improvement of the manuscript:

I would strongly recommend to use word ‘knock-down’ instead of ‘silencing’ (lines 133, 138, 143, 152, 160, 180, 182, 184, 226, 230) and definitely not to use ‘loss of function’(line 34) describing knock-down experiments especially when knock-down efficiency hardly exceeds 60%.  

Line 28, … was [found] significantly upregulated…

Lines 30- 31, 103, … high [level of] PANTR1 expression… The reduction to ‘high/low expression’ is suitable for figure labels but not in the main text.

Line 40, …[statistically] significant positive correlation…

Lines 56-59, Grammar. Re-write or better make two sentences.  

Line 67, … range of [lncRNA] functions…  

Line 69, … Aberrant functions [of lncRNAs]…

Line 79, add ‘studied’

Line 80, delete ‘studied’

Line 121, It is not correct to call PANTR1 hybridization signals ‘positive stainig’

Lines 124, 125, 213, the right spelling is clinicopathological

Line 132, I don’t think the authors of this paper ‘established knock-down approach’, they rather applied already well-established protocol to knock-down PANTR1 lncRNA. And based on method description, they did not even work much on optimization.

Line 141, cell death control siRNA cannot serve as a transfection efficiency control. However, it is an essential positive control for cell growth reduction.

Line 149, use knocked-down instead of silenced

Lines 147-150, Re-write the sentence. I would include references to original papers on PARP cleavage as an indicator of caspase activity and apoptosis, e.g. papers of the Poirier group from the 90s, alternatively a reference to a recent review that describes methods to detect apoptosis will work.  

Lines 157-158, It is not clear what bands were analyzed and what is ‘the control band’. Please extend.

Line 158, housekeeping protein not ‘housekeeper’

Lines 163-164, you can skip ‘of PANTR1’ first time after ‘expression’ but do not omit it after ‘siRNA-mediated knock-down’.

Lines 165-166. Grammar issues

Line 191, involvement (typo)

Line 292, Use ‘hybridized’ instead of ‘stained’

Lines 308, 370. What 50 nM siRNA concentration stands for is not clear. Was it a final concentration of siRNA after adding complexes to cells? What was the rationale for using this very high concentration?

Lines 335-350. What instrument and software were used to image WBs? Is the system compatible with quantitative analysis?

Figure 1.

I did not get why panels c and d of figure 1 show plots for normal and carcinoma tissues in different order. Figure 1 legend contains confusing abbreviations: (line 116) GB should be GBM, and (line 117) KI should be KIRC to match panel b of the figure and line 91 of Results.

By the way, in line 91 KIRC is better to introduce after ccRCC, which was spelled out above, while KIRC is a new acronym for a reader.

Figure 3. Please indicate corresponding cell lines on WB panels. Add 'kDa' to 89 and 116 in parentheses.

Suppl. Figure S1.

I wonder if there are any statistically significant differences in PANTR1 expression levels between these cell lines.

Suppl. Figures S2 and S4. What time point after siRNA transfection was chosen to check knock-down efficiency?

Author Response

Response to Reviewer 2

This manuscript represents a simple and straightforward study of potential roles of non-coding RNAs in cancer progression. Specifically, the study aimed to identify a molecular marker associated with clear-cell renal cell cancer prognosis. The authors claim that they identified such a marker, lncRNA PANTR1. The field is still in a stage of data accumulation, so the more markers are found the better our chances to develop new targeted cancer treatment in future.

Reviewer #2: Overall, it is a reasonably well-written manuscript. Yet, it has some grammar issues and wrong wording throughout the text. Discussion section in its present form is a rephrased version of Results. If the authors have very little to discuss they should consider combining Results and Discussion.

Authors reply: The discussion has been strengthened and extended and is hopefully not anymore, a rephrased version of the Results.

Reviewer #2: My major concern is that it seems the authors overstate the significance of some of their findings. In abstract (line 40) and Results (line 177) they are talking about ‘significant correlation’. What is this significant correlation? Correlation could be weak or strong, positive or negative. The calculated in this study correlation coefficients R=0.19 and 0.13, while being statistically significant, indicate very weak positive correlation.

Authors reply: We are grateful for this important reviewer´s comment and have now adapted our statements in the abstract and results section refraining from using the terms statistically significant (lines 42-43 and 186) and stating now that the correlation is positive but weak (line 186).

Reviewer #2: Effects of PANTR1 knock-down on cell growth and caspase activity are rather minor and somewhat inconsistent (Figures 2 and 3). The fact that changes were statistically significant, does not make the changes prominent.

Authors reply: We fully agree with the constructive reviewer’s opinion that the reported effects are in some cell lines rather moderate (though statistically significant), but based on the observation and confirmation in three independent cell lines, we are convinced that this effect is biological and reproducible in independent cell lines. One reason for the varying size effect, might be the situation of genetic and molecular heterogeneity of different cancer cells, as we frequently also encounter the same issue in our patients. For instance, about a third of patients are responding to multi-kinase anti-angiogenic inhibitor drugs, about 40 to 60% are responding to immunotherapy, respectively. We included this point of view in the discussion section of the manuscript (lines 254-258).

Reviewer #2: Some minor points and suggestions for further improvement of the manuscript:

I would strongly recommend to use word ‘knock-down’ instead of ‘silencing’ (lines 133, 138, 143, 152, 160, 180, 182, 184, 226, 230) and definitely not to use ‘loss of function’(line 34) describing knock-down experiments especially when knock-down efficiency hardly exceeds 60%.  

Line 28, … was [found] significantly upregulated…

Lines 30- 31, 103, … high [level of] PANTR1 expression… The reduction to ‘high/low expression’ is suitable for figure labels but not in the main text.

Line 40, …[statistically] significant positive correlation…

Lines 56-59, Grammar. Re-write or better make two sentences.  

Line 67, … range of [lncRNA] functions…  

Line 69, … Aberrant functions [of lncRNAs]…

Line 79, add ‘studied’

Line 80, delete ‘studied’

Line 121, It is not correct to call PANTR1 hybridization signals ‘positive staining’

Lines 124, 125, 213, the right spelling is clinicopathological

Line 132, I don’t think the authors of this paper ‘established knock-down approach’, they rather applied already well-established protocol to knock-down PANTR1 lncRNA. And based on method description, they did not even work much on optimization.

Line 141, cell death control siRNA cannot serve as a transfection efficiency control. However, it is an essential positive control for cell growth reduction.

Line 149, use knocked-down instead of silenced

Lines 147-150, Re-write the sentence. I would include references to original papers on PARP cleavage as an indicator of caspase activity and apoptosis, e.g. papers of the Poirier group from the 90s, alternatively a reference to a recent review that describes methods to detect apoptosis will work.  

Lines 157-158, It is not clear what bands were analyzed and what is ‘the control band’. Please extend.

Line 158, housekeeping protein not ‘housekeeper’

Lines 163-164, you can skip ‘of PANTR1’ first time after ‘expression’ but do not omit it after ‘siRNA-mediated knock-down’.

Lines 165-166. Grammar issues

Line 191, involvement (typo)

Line 292, Use ‘hybridized’ instead of ‘stained’

Authors reply: All suggested grammatical, writing and citation changes have been adapted.

Reviewer #2: Lines 308, 370. What 50 nM siRNA concentration stands for is not clear. Was it a final concentration of siRNA after adding complexes to cells? What was the rationale for using this very high concentration?

Authors reply: We used a final concentration of 50 nM siRNA – this is the concentration finally in the wells of the cells; this information has been added to materials and methods (lines 361, 407, 423). Usually, whereas mRNAs of protein-coding genes need concentrations around 10 nM siRNAs, lncRNAs commonly need higher concentration of siRNA to get efficient knock-down effects, and 50 nM is a concentration many times used in other published studies of our own and other groups.

Reviewer #2: Lines 335-350. What instrument and software were used to image WBs? Is the system compatible with quantitative analysis?

Authors reply: For image acquisition ChemiDocTM Touch (Biorad) was used and image analysis and quantification were performed with ImageJ. We have included this information in the material and methods section (lines 402-403).

Reviewer #2: Figure 1 - I did not get why panels c and d of figure 1 show plots for normal and carcinoma tissues in different order. Figure 1 legend contains confusing abbreviations: (line 116) GB should be GBM, and (line 117) KI should be KIRC to match panel b of the figure and line 91 of Results. By the way, in line 91 KIRC is better to introduce after ccRCC, which was spelled out above, while KIRC is a new acronym for a reader. 

Authors reply: We thank the reviewer for carefully reading these confusing labelling. The points addressed by the reviewer have been corrected and the labelling has been changed in Figure 1 as well as in the figure legend and text (lines 94 and 125-126).

Reviewer #2: Figure 3. Please indicate corresponding cell lines on WB panels. Add 'kDa' to 89 and 116 in parentheses.

Authors reply: Changes have been made in Figure 3.  

Reviewer #2: Suppl. Figure S1. I wonder if there are any statistically significant differences in PANTR1 expression levels between these cell lines.

Authors reply: The levels of PANTR1 expression between ccRCC cell lines and HUVECs differs significantly, statistical analysis has been included in Figure S1.

Reviewer #2: Suppl. Figures S2 and S4. What time point after siRNA transfection was chosen to check knock-down efficiency?

Authors reply: Knock-down efficiency was assessed 48 h after transfection. The information has been added to the figure legends.

Reviewer 3 Report

In this paper the Authors analyse the role of PANTR1, an oncogenic long non-coding RNA and attribute it a role in renal cell carcinoma (RCC). Specifically, they find that PANTR1 is significantly up-regulated in clear-cell renal cell carcinoma tissue (ccRCC) compared to non-cancerous kidney tissue and that its expression results in poor disease-free survival. To analyse the role of the lncRNA, they verify the effects of its downregulation by siRNAs. A decrease in PANTR1 levels reduces cellular growth, activates apoptosis, stimulates in vitro tube formation and endothelial cell migration. At the molecular level, its expression is related to VEGF-A and LAMC2 transcription. In the present form, the manuscript suffers of some limitation. I suggest to deal with the following implementations.

  • The Authors show that PANTR1 is expressed in KIRC and KIRP tumours but not in KICH, where it is expressed in normal samples. This aspect should be explained and the putative role of the lncRNA in this context better addressed.
  • In fig. 2 the Authors analyse cell proliferation and claim that PANTR1 depletion reduces cell growth in the cell lines they utilize. In my opinion this is an overstatement and the data should be shown in a different way (tune down the conclusion) or at least the reported data should be limited to sensible cell lines.
  • The expression of the lncRNA seems to modulate the expression levels of different genes. On the basis of literature data reporting the epigenetic effects of lncRNAs in gene expression regulation, the Authors should discuss/propose/demonstrate the mechanism by which lncRNA PANTR1 regulates its targets.

Author Response

Response to Reviewer 3

Comments and Suggestions for Authors

In this paper the Authors analyse the role of PANTR1, an oncogenic long non-coding RNA and attribute it a role in renal cell carcinoma (RCC). Specifically, they find that PANTR1 is significantly up-regulated in clear-cell renal cell carcinoma tissue (ccRCC) compared to non-cancerous kidney tissue and that its expression results in poor disease-free survival. To analyse the role of the lncRNA, they verify the effects of its downregulation by siRNAs. A decrease in PANTR1 levels reduces cellular growth, activates apoptosis, stimulates in vitro tube formation and endothelial cell migration. At the molecular level, its expression is related to VEGF-A and LAMC2 transcription. In the present form, the manuscript suffers of some limitation. I suggest to deal with the following implementations.

Reviewer #3: The Authors show that PANTR1 is expressed in KIRC and KIRP tumours but not in KICH, where it is expressed in normal samples. This aspect should be explained and the putative role of the lncRNA in this context better addressed.

Author' s reply: We are grateful for this comment and indicating to this interesting point. Though we do not have an experimental explanation, one can speculate about the differences in the underlying pathophysiology of renal cell carcinoma (RCC). For instance, in the most common renal cell carcinoma subtype, the clear cell RCC angiogenesis is a major activated hallmark through inactivation of von-Hippel-Lindau protein/overactivation of HIF2 pathway. The papillary RCC is a rather heterogenous group, including genetic alterations in cMet signaling and metabolic enzymes. In chromophobe RCC, alterations in Birt-Hogg-Dube´are commonly found. Thus given this genetic and molecular heterogeneity of RCC, that might explain why PANTR1 is differentially expressed in different subtypes of RCC (lines 219-236).

Reviewer #3: In fig. 2 the Authors analyze cell proliferation and claim that PANTR1 depletion reduces cell growth in the cell lines they utilize. In my opinion this is an overstatement and the data should be shown in a different way (tune down the conclusion) or at least the reported data should be limited to sensible cell lines.

Authors reply: We fully agree with the reviewer’s opinion that the reported effects are in some cell lines rather moderate (though statistically significant), but based on the observation and confirmation in three independent cell lines, we believe that this effect is true and reproducible. More importantly, this also reflects the genetic and molecular heterogeneity of different cancer cells, as we frequently encounter the same issue in our patients. For instance, only a third of patients are responding to multi-kinase anti-angiogenic inhibitor drugs, about 40 to 60% are responding to immunotherapy, respectively. We included this point of view in the discussion of the manuscript (lines 254-258).

Reviewer #3: The expression of the lncRNA seems to modulate the expression levels of different genes. On the basis of literature data reporting the epigenetic effects of lncRNAs in gene expression regulation, the Authors should discuss/propose/demonstrate the mechanism by which lncRNA PANTR1 regulates its targets.

Authors reply: This is a very good statement by the reviewer. Based on this, a new paragraph about the role of lncRNAs in epigenetic regulation of gene expression in cancer has been added to the discussion section. The general mechanisms of epigenetic lncRNA regulation, as well as representative examples, the role of PANTR1 in epigenetic regulation and the corresponding citations have been added (lines 283-298).

Round 2

Reviewer 3 Report

In my opinion the paper can be accepted in the present form